# Modeling of DNA Damage Repair and Cell Response in Relation to p53 System Exposed to Ionizing Radiation

**DOI:** 10.3390/ijms231911323

**Published:** 2022-09-26

**Authors:** Ankang Hu, Wanyi Zhou, Zhen Wu, Hui Zhang, Junli Li, Rui Qiu

**Affiliations:** 1Department of Engineering Physics, Tsinghua University, Beijing 100084, China; 2Key Laboratory of Particle & Radiation Imaging, Tsinghua University, Ministry of Education, Beijing 100084, China; 3Nuctech Company Limited, Beijing 100084, China

**Keywords:** track structure, Monte Carlo, DNA repair, p53, ionizing radiation

## Abstract

Repair of DNA damage induced by ionizing radiation plays an important role in the cell response to ionizing radiation. Radiation-induced DNA damage also activates the p53 system, which determines the fate of cells. The kinetics of repair, which is affected by the cell itself and the complexity of DNA damage, influences the cell response and fate via affecting the p53 system. To mechanistically study the influences of the cell response to different LET radiations, we introduce a new repair module and a p53 system model with NASIC, a Monte Carlo track structure code. The factors determining the kinetics of the double-strand break (DSB) repair are modeled, including the chromosome environment and complexity of DSB. The kinetics of DSB repair is modeled considering the resection-dependent and resection-independent compartments. The p53 system is modeled by simulating the interactions among genes and proteins. With this model, the cell responses to low- and high-LET irradiation are simulated, respectively. It is found that the kinetics of DSB repair greatly affects the cell fate and later biological effects. A large number of DSBs and a slow repair process lead to severe biological consequences. High-LET radiation induces more complex DSBs, which can be repaired by slow processes, subsequently resulting in a longer cycle arrest and, furthermore, apoptosis and more secreting of TGFβ. The Monte Carlo track structure simulation with a more realistic repair module and the p53 system model developed in this study can expand the functions of the NASIC code in simulating mechanical radiobiological effects.

## 1. Introduction

Ionizing radiation results in DNA damage, among which the double-strand break (DSB) is the most serious one. Repair of DSBs induced by ionizing radiation is a critical process of radiobiological effect. Besides the results of the repair, the kinetics of repair also affects the fate of cells [1]. For a cell in the G2 phase, nonhomologous end joining (NHEJ) and homologous recombination (HR) are two main pathways for repairing DSBs [2]. For a cell in the G0/G1 phase, NHEJ is the main pathway of DSB repair. For the repair curves of NHEJ in G0/G1, there are two compartments of NHEJ classified by different processes: the resection-independent compartment and the resection-dependent compartment [3]. Resection-independent NHEJ is a relatively fast and simple process, and the DSB is repaired without resection [1]. Several proteins are related to the compartment, such as Ku70/80, DNA-PKcs, Ligase IV, and XRCC4 [4]. The resection-dependent compartment is a slow and complex process involving the resection of DNA ends and many types of proteins, such as CtIP, EXO1, and Artemis [5,6]. The scheme of the two compartments in NHEJ repair is shown in Figure 1. Many factors affect the pathway choice of the NHEJ in G0/G1 cells, such as the chromatin environment and the complexity of DSBs [6,7,8]. The choice of a model for the repair pathway is essential for radiobiological study because of the critical influence of repair kinetics on the fate of cells [9,10].

In addition to DNA damage, ionizing radiation also activates the signaling system of cell response to DNA damage [11]. The tumor suppressor p53 and related systems play important roles in the response system by regulating DNA repair, cell cycle suppression, and the initiation of apoptosis [11,12]. The function of p53 is complex and controls the expression of numerous genes of contradictory roles: prosurvival, cell cycle suppressing, or pro-apoptosis [13]. A sophisticated system centered on p53 is constructed by related proteins [13]. The system of p53 consists of several feedback loops [14,15], which means that the kinetics of the external factors that activate the p53 system also influence the p53 system [16]. Besides the number of DNA lesions, the p53 system is influenced by the dynamics of DNA lesions [16]. Compared with low-LET radiation, high-LET radiation induces a larger number of complex DSBs [9,17]. Because of different fractions of complex DSBs, the kinetics of DSB repair differ greatly between low- and high-LET radiation [17], which influences the p53 system, leading to different results. An integrated study of DNA damage repair and the p53 system can be used to explore the mechanism of cell response to low- and high-LET radiation.

Monte Carlo track structure codes are useful tools for the mechanical research of radiobiological effects on different conditions [18]. These codes can be used to simulate the interactions between radiation and target in the cell, such as DNA, and obtain information on lesions, such as position, type, and the track inducing the lesion [18,19,20]. The process of DNA repair can be simulated mechanically according to the information of lesions. Stochastic methods [21,22] and deterministic methods [1,4,23,24] are used in the simulation of DNA repair. Several models consider the resection-independent and resection-dependent NHEJ [22,25]. A Monte Carlo track structure code combined with the repair model provides a mechanistic simulating chain to study the DNA damage, repair, and its consequence induced by ionizing radiation. However, the fractions of repair compartments in these models are set to fixed values, which limits the use of the model for high-LET radiation.

Modeling of the p53 system has been studied in the fields of computational biology and systems biology. The main aim of modeling the p53 system is to explore how the related factors influence cell response and cell fate by taking into account the complexity and uncertainty of the cellular system. Several models have been constructed to study the features of p53 feedback loops [14], the fate of cell fate decision-making by the p53 system [16,26], and the response to DNA damage related to p53 [11,15]. An integrated model containing the track structure code, mechanistic repair module, and p53 system model can expand the mechanistic study of radiobiology to further and enhance research fields [12].

Our lab has developed a Monte Carlo track structure code NASIC (Nanodosimetry Simulation Code) to simulate the radiobiological effects [27,28]. The code contains the physical, physicochemical, chemical, damage, and simplified repair modules. In this work, we extend the NASIC code by adding a new repair module and the p53 system model, which can fill the gaps in the complete chain of the mechanical simulation of the radiation-induced biological effects. To our best knowledge, it is the first time to consider the effect of complex DSB on DNA repair kinetics and introduce the p53 system into the radiobiological simulation process. Furthermore, the NASIC code implementing these modules can be used to study the regulations of cell response to radiotherapy related to the p53 system by considering cell factors, chemotherapeutic agents, and other factors [26]. This advanced code further provides a tool to mechanically explore the biological effects of low- and high-LET radiation.

## 2. Results and Discussion

### 2.1. Influence of DNA Damage Complexity on the Fractions of Slow and Fast DNA Repair

As mentioned above, the chromatin environment and the complexity of DSBs are two factors affecting the kinetics of DSB repair. As the chromatin environment has been proven to be a factor influencing the fraction of resection-dependent repair [8], we focus on the influence of the complexity of DSBs on the repair kinetics. We fitted the fraction of the slow compartment using the repair curves in experiments with high-LET radiation [17,29,30,31]. The data of unrepaired DSBs at 24 h after α particle irradiation for the Artemis-deficient cells are also used to represent the fraction of the resection-dependent compartment [32]. Then, the fractions are processed according to Equation (2) to calculate the ExF(radiation) values for each experiment. We also simulate the DNA damage for each condition and calculate the DSB_c_ to represent the complexity of DSBs. The ExF(radiation) seems linearly related to the fraction of DSB_c_. The Pearson correlation coefficient is 0.764 (*p* = 0.01). The data and the fitting line are shown in Figure 2. The radiation conditions of the experiments are also marked in Figure 2.

The equation of the fitting line is shown in Equation (1). The R-square of the fitting line is 0.58.


(1)
$${\mathrm{ExF}}_{\mathrm{slow}}=0.8268{\mathrm{DSB}}_{c}+0.0705$$


The relationship between DSB_c_ and ExF_slow_ indicates that additional strand breaks near the DSB would lead to repair by the slow compartment. The linear relationship supports our model of factors regulating slow and fast DNA repair. The simple DSBs located in euchromatin are repaired by a simple resection-independent compartment. The additional strand breaks near the DSBs make the simple resection-independent repair fail, thus resulting in the resection-dependent compartment. Biehs et al. [5] proposed a model that Ku70/80 moves away from the break ends to expose DNA ends for nuclease access while limiting the extent of resected DNA. Then, EXO1, Mre11, and EXO2 resect the nucleotides between the end and Ku70/80. The reporter assay by Biehs et al. revealed that resection-dependent end joining is associated with nucleotide losses of 5–20 bp [5]. It seems to indicate that the resection removes the strand breaks near the DSB for subsequent repair. The model and data of nucleotide losses by Biehs et al. also reveal the rationality of defining the complex DSB by additional strand breaks near the DSB.

According to the general and hypothetical model of DNA repair compartments, low-LET radiation induces simple DSBs mainly repaired in the fast compartment. On the contrary, high-LET radiation induces more complex DSBs so that more slow repairs are required. High-LET radiation affects not only the repair results, such as the fraction of misrepair, but also the repair kinetics. The different repair kinetics of DSBs may affect the fate of cells via a complex system in the cell.

It should be noted that there are non-negligible uncertainties in the fractions obtained by fitting the repair curves. Moreover, the DSB_c_ is calculated by the Monte Carlo simulation rather than the measurement from experiments. The Monte Carlo simulation may lead to systematic bias because of ignoring factors in the physical and chemical simulation and the nucleus models. The uncertainty of the fractions from repair curves and the systematic uncertainties in the Monte Carlo simulation might suggest that the Pearson correlation coefficient and fitting line in this work do not reflect the actual situations of the DNA repair.

### 2.2. Kinetics of DNA Repair

In this study, the fraction of the slow repair compartment is changed to simulate the dynamics of the repair for different conditions. Repair curves are compared with the models of Friedland et al. [21] and Taleei et al. [33] in Figure 3.

Our results show a similar repair rate to the repair curve of Taleei et al., and the fraction of the slow compartment in their model is between 10% and 20%. Taleei et al. used the data of X-ray experiments, according to which the fraction of the slow compartment after X-ray irradiation is 15–20%. The match between our simulation and the experiment shows that our DNA repair model is reasonable.

The curve of Friedland et al. shows a faster repair rate than that of Taleei et al. for the beginning period of the repair. It could be estimated by the curve that the fraction of slow compartments is between 30% and 40%, while the slow and fast compartments were not modeled separately in their study. Friedland et al. proposed that the slow phase of repair curves is due to the interference of the damage near the DSB, which may lead to the difference that they did not introduce the resection-dependent compartment in their model to describe. Their parameters were obtained by fitting data containing the experiments of high-LET radiation, which may explain their higher fraction of the slow compartment.

It should be pointed out that the details of the DNA repair remain unclear. Thus, many repair-related proteins are ignored by the model. The parameters in this model are obtained by limited experiments. These parameters are influenced by the type and state of the cells and many other factors. The differences between models also indicate that the model is limited to specific conditions.

The repair curves show that the cells irradiated by high-LET radiation contain a high level of unrepaired DSBs for a long time. It results in a higher probability of misrepair and longtime activation of the DNA repair-related system.

### 2.3. Kinetics of p53 System and Stochastic Simulation of Cell Fate

Trajectories of model components in the p53 system are simulated by the deterministic simulation. The conditions of 2 Gy and 8 Gy irradiation with slow compartments of 20% and 60% are shown in Figure 4 as an example.

The amplitudes of ATM_p_ and phosphorylated p53 are similar for cells exposed to 2 Gy and 8 Gy because the activation of ATM by DSBs performs similar to a Hill mechanism so that the activation tends to saturate for a large number of DSBs [15]. The trajectories also show oscillations of phosphorylated p53, Mdm2, and Wip1, with a period of around eight hours. The oscillations are attributed to the negative feedback loops of ATM, p53, Mdm2, and Wip1. The curves of cyclin E tending to be zero indicates cell cycle arrest. The number of Bax is related to apoptosis. The simulated result shows that a longer time of cycle arrest and more apoptosis is attributed to the large number of DSBs induced by a high dose, and the long time of repair results from a large fraction of slow compartments. The high-LET irradiation induces more complex DSBs, which should be repaired by a slow compartment, leading to more severe cycle arrest and apoptosis.

The secreting of TGFβ is also simulated for conditions with doses ranging from 2 Gy to 8 Gy and fractions of the slow repair compartment ranging from 20% to 80%. The results are shown in Figure 5.

The pathway of p21-GADD45-p38-TGFβ is one of the pathways regulating the secreting of TGFβ after exposure to irradiation. The cells irradiated by a high dose and repaired for a long period would activate the p53 system for a long time, activating p21 and subsequent TGFβ to a greater extent. TGFβ is an important factor related to fibrosis. It is predicted that the cells repaired by a slower process would result in more severe fibrosis compared with the fast-repaired cell irradiated by the same dose.

A stochastic simulation is performed to calculate cell apoptosis for different conditions. Besides the dose and the fraction of the slow compartment, we consider the parameters of ATM activation by DSB. The results of the apoptosis percentages at 72 h after irradiation are shown in Figure 6 for different parameters of ATM activation, doses, and fractions of the slow repair compartment.

Results show that not only the dose but also the repair kinetics affect apoptosis. Moreover, the parameter of ATM activation has a greater impact on cell fate.

ATM activation reaches saturation after low-dose irradiation. The negative feedback loops in the p53 system result in limited response to the DNA damage. These two factors determine that the last time of DNA damage affects the response of the p53-related system more significantly than the number of lesions in radiation exposure. The longer last time of lesions leads to continuous oscillation of the p53 system, which results in more severe consequences, including cell cycle arrest, apoptosis, and secreting of TGFβ. Then, these effects influence immune reactions, tissue repair, and fibrosis. High-LET radiation induces more complex lesions. The repair of these complex lesions requires longer, leading to more serious consequences than the same number of lesions induced by low-LET radiation.

The features of the p53 system also indicate that repair kinetics is an important part of modeling the cell response to ionizing radiation. The focus should not be limited to the results of DNA repair, such as chromosome aberration and unrepaired DSBs. The process of DSB repair also plays an essential role in the radiobiological effect.

The p53 system introduced in this work also provides a tool for researching the factors regulating the DNA repair-related system. The influence of drugs can be simulated by regulating the gene, mRNA, or proteins as the target of the drug. The p53 system in this work only considers the main and dominant part of the system related to DNA repair. More efforts should be made to complete the system in a future study.

### 2.4. Limitations

In the model, determining the fraction of the slow and fast compartments generally takes two factors into account, the chromatin environment and the complexity of DSBs. The detailed mechanism requires further research. The data about the complexity of DSBs are obtained by Monte Carlo simulation but not experimental data. The data from the repair curves are highly uncertain, and there are divergences between experiments. The results of this study indicate that the slow compartment repairs the DSBs accompanied by additional breaks. More targeted experiments are needed to explore how complex DSBs affect the kinetics of repair. The complex process of the slow (resection-dependent) repair remains unclear [6]. The model in this study is simplified with limited experimental data. Actually, the model of the p53 system is not aimed to directly compare the simulation results with experiments because of the complexity of the biological system and the variety of the cells. This study aims to provide a scientific framework for exploring specific factors’ influence on the dynamic changes in mRNAs and proteins in the system and the subsequent fate of cells.

## 3. Materials and Methods

### 3.1. Overview of NASIC

In this study, we expand the NASIC by adding a new repair module and p53 system modules. With the original modules of NASIC and extended modules, we can simulate the kinetics of DSB repair, the time-dependent change in proteins in the p53 system, and the fate of cells after exposure to low- and high-LET radiation. The DSBs information is obtained by the track structure simulation (physical, physicochemical, chemical, and damage modules in NASIC). Then, the damage data are processed as the input of the repair module and p53 system module. The repair module and p53 system module simulate the cell response to damage. The simulation chain with these modules is used to explore how the low- and high-LET radiation affects the biological effect by influencing the kinetics of DSB repair. The scheme of NASIC with the new modules is shown in Figure 7.

### 3.2. DNA Damage Simulation

DNA damage is simulated by NASIC [27,28] with an atomic nucleus model at the G0/G1 phase. The model describes different orders of chromosome packing structures and the coordinates of atoms in a deoxyribonucleotide [27]. Particles are set and simulated according to the type and energy around the nucleus model to study the effects of low- and high-LET radiation. The physical module simulates the interactions between radiation and the nucleus. The radiolysis products are simulated and added to the code by the physicochemical module. The chemical module calculates reactions and lesions induced by free radicals from radiolysis products. Then, direct damage and indirect damage are gathered to an output file of NASIC, recording the location, damage type, and track information of lesions. The output file is used as the input of the subsequent new modules.

### 3.3. Modeling of Factors Regulating Slow and Fast DNA Repair

NHEJ is the main pathway for repairing DSBs in G0/G1 phase cells. Chromatin environment and damage complexity are two important factors affecting the pathway of NHEJ [6,32]. If a DSB locates in heterochromatin, it requires ATM signaling to phosphorylate Kap-1 for chromatin relaxation and subsequent access to repair proteins, and it is primarily repaired by slow NHEJ [8]. For the DSBs in euchromatin, the type of NHEJ is highly related to the complexity of DSBs: Complex DSBs are repaired by slow NHEJ [5,6,9,34], while fast NHEJ repairs simple ones. Therefore, we establish a model considering two factors, chromatin environment and complexity of DSBs, regulating slow and fast NHEJ repair model. The scheme of the model is shown in Figure 8.

In this model, the fraction of slow NHEJ repair of DSBs induced by X-ray and other types of radiation can be defined in Equation (2).


(2)
$$F_{\mathrm{slow}}=\mathrm{HC}+(1-\mathrm{HC}){\mathrm{DSB}}_{\mathrm{complex},\mathrm{EC}}(\mathrm{radiation})$$


*HC* is the fraction of DSBs in heterochromatin; *DSB_complex,EC_* is the fraction of complex DSBs in euchromatin. The *DSB_complex,EC_* is determined by the radiation. Complex DSBs often refer to the clustering of several different DNA lesions within a short DNA region of 10–15 bp [35]. Complex DSBs are not accurately defined, and their detection is challenging in biological experiments. Therefore, we use the results obtained by track structure Monte Carlo simulation to establish the model of choice for the repair pathways. The DSBs are classified by the lesions within 10 bp of the DSB according to the method of Nikjoo et al. [36] and Stewart et al. [37]. Two types of DSBs are considered: DSB_0_ are double-strand breaks accompanied by zero lesion on a strand within 10 bp; DSB_c_ are double-strand breaks accompanied by one (or more) additional break(s) on a strand within 10 bp.

To build the relationship between the complexity of DSB and the fraction of the slow repair, we analyze the repair kinetics from experiments and yields of complex DSBs from the simulation. The fractions of slow and fast repair are obtained by fitting the repair curves of DSBs from experiments in the literature. These experiments cover the cells irradiated by X-ray [17], proton [38], helium ions [29,32], carbon ion [17], nitrogen ion [29], neon ion [30], and iron ion [30,31]. The type and energy of the particles for each condition are marked in the Figure 2 in Section 2.1. The yields of complex DSBs are calculated by setting the same radiation condition in NASIC as the experiments in the literature. The fractions of heterochromatin may differ for different types of cells [39]. To avoid the influence of this difference on data from different experiments, we introduce a new quantity, the extra fraction of slow repair compared to X-ray (ExF_slow_), in Equation (3).


(3)
$$\begin{array}{l} {\mathrm{ExF}}_{\mathrm{slow}}(\mathrm{radiation})=\frac{F_{\mathrm{slow}}(\mathrm{radiation})-F_{\mathrm{slow}}(X-\mathrm{ray})}{1-F_{\mathrm{slow}}(X-\mathrm{ray})}\\ =\frac{\left(1-\mathrm{HC})({\mathrm{DSB}}_{\mathrm{complex},\mathrm{EC}}(\mathrm{radiation})-{\mathrm{DSB}}_{\mathrm{complex},\mathrm{EC}}(X-\mathrm{ray})\right)}{\left(1-\mathrm{HC})(1-{\mathrm{DSB}}_{\mathrm{complex},\mathrm{EC}}(X-\mathrm{ray})\right)}\\ =\frac{{\mathrm{DSB}}_{\mathrm{complex},\mathrm{EC}}(\mathrm{radiation})-{\mathrm{DSB}}_{\mathrm{complex},\mathrm{EC}}(X-\mathrm{ray})}{1-{\mathrm{DSB}}_{\mathrm{complex},\mathrm{EC}}(X-\mathrm{ray})} \end{array}$$


*F_slow_*(*X-ray*) and *F_slow_*(*radiation*) are fractions of slow repair by X-ray and high-LET radiations in the same experiments. ExF_slow_(*radiation*) reflects the influence of complex DSBs by different LET radiation on repair kinetics.

### 3.4. DNA Repair Model

The new DNA repair model is developed on the basis of the simplified repair module of NASIC [28]. The simplified repair module in the original version of NASIC can only be used to simulate fast NHEJ repair. The slow repair-related process has been added to this new repair model. Because some experiments show that most DSBs induced by high-LET radiation are repaired with similar kinetics to the X-ray-induced slow component, and the fraction of the slow compartment is dose-independent [6], we use the same kinetics to simulate the slow compartments due to heterochromatin and complex DSBs.

The process of DSB repair is modeled by considering the change in the state of the DSB ends in a step-by-step manner. In this process, repair-related proteins are recruited to the DSB end to change the state of the end. The probability of state change in the step is calculated according to the kinetics of protein recruitments. Meanwhile, the end walks randomly during the step in a manner described by Friedland et al. [21].

In this model, DSBs are classified into two types that are repaired by fast (resection-independent) or slow (resection-dependent) pathways according to the chromatin environment and the complexity of the DSBs [3]. The chromatin environment of each DSB is determined by randomly sampling according to the fractions of heterochromatin and euchromatin because of the lack of chromatin environment information in the nucleus model. For both pathways, the Ku70/80 heterodimers bind to the end and serve as platforms for loading the NHEJ-related protein. Then, DNA-PKcs is recruited to the end. DNA-PKcs and Ku form the DNA-PK complex. When two DSB ends with attached DNA-PK complexes close to each other, a synaptic complex is formed. After the synapsis, DNA-PK autophosphorylates the DNA-PKcs protein for subsequent recruitment of the proteins [4,21]. For the fast repair pathway, Ligase Ⅳ, XRRC4, and other enzymes are recruited to the end and repair the end in this model. As for the slow repair pathway, the DNA ends are resected and processed by CtIP, EXO1, and Artemis. Then, the processed ends are added nucleotides by polymerase μ or λ before the ligation process. Finally, the DSBs are repaired by ligase Ⅳ, XRRC4, and other enzymes. The recruitment of CtIP and EXO1 starts after the binding of Ku70/80, and Artemis is recruited with DNA-PKcs [22]. The scheme of the DNA repair model with fast and slow compartments is shown in Figure 9.

The parameters of the kinetics of protein recruitment, synapsis, and random walk of the end are listed in Table 1 according to the parameters from Friedland et al. [21], Taleei et al. [4], and Qi et al. [22]. The parameters k_1_–k_12_ are the rate constants of protein recruitment. The k is the reciprocal of the characteristic time of the recruitment. If the distance between two DNA ends is less than a value, a synaptic complex is formed. The value is defined as synapsis distance. The diffusion coefficients of DNA ends are obtained from Friedland et al. [21].

**Table 1 ijms-23-11323-t001:** Parameters of the DNA repair model.

Parameter	Value	Reference
k_1_	1/0.1 s^−1^	[21]
k_2_	1/3.0 s^−1^	[21]
k_3_	1/72.0 s^−1^	[4]
k_4_	1/180.0 s^−1^	[4]
k_5_	1/240.0 s^−1^	[4]
k_6_	1/720.0 s^−1^	[4]
k_7_	1/7.0 s^−1^	[22]
k_8_	1/1.2 s^−1^	[22]
k_9_	1/1000.0 s^−1^	[4]
k_10_	1/450.0 s^−1^	[4]
k_11_	1/14,400.0 s^−1^	[4]
k_12_	1/6545 s^−1^	[4]
Synapsis distance	100 nm	[21]
Diffusion coefficient of DNA end before synapsis	1.0 nm^2^/s	[21]
Diffusion coefficient of DNA end after synapsis	100.0 nm^2^/s	[21]

### 3.5. P53 System

The p53 system module of NASIC is developed on the basis of the model in Hat et al. [26] and Bogdal et al. [40] in this work. The module consists of three parts: (1) the feedback network of the p53, (2) the cell cycle arrest network regulated via p21, and (3) the apoptosis network regulated by Bax and phosphorylated Akt (Akt_p_). In this study, the number of DSBs is obtained from the repair module of NASIC rather than the simplified kinetics modeling of Hat et al. [26]. The repair module and the p53 system module execute simultaneously when simulating the cell response to radiation so that we can explore how the repair kinetics affects the dynamic of proteins and the fate of cells. Moreover, we introduce the pathway of TGFβ through p21-GADD45-p38(MAPK14)-TGFβ. TGFβ is an important factor related to cancer progression, immunology, and fibrosis [41,42,43]. With this module, the effects of radiation on immunology and fibrosis can be studied. The scheme of the p53 system is shown in Figure 10.

In this system, radiation induces DSBs, leading to the activation of ATM via intermolecular autophosphorylation [44]. The activation of ATM by DSBs performs similar to a Hill mechanism [15]. The number of activated ATMs increases with the radiation dose and tends to saturate to a fixed value [15]. Then, activated ATM stabilizes and activates p53 by phosphorylating it to one of its active phosphoforms (at Ser15 and Ser20): p53 arrester. This p53 arrester phosphofrom can promote cell cycle arrest and the secreting of TGFβ via the p21 signal [16,45]. This phosphoform can be further phosphorylated (at Ser46) to the p53 killer form. The form promotes apoptosis attributed to Caspases via the Bax signal [26]. Meanwhile, the p53 arrester induces the synthesis of Mdm2 and Wip1 to inhibit ATM, the p53 killer, and the p53 arrester. These interactions construct a negative feedback loop. Detailed descriptions of the p53 system and the parameters used in the simulation are listed in the Appendix A. The interactions between proteins, mRNA, or genes are modeled and described as chemical reaction formulas. The rate constants of these interactions are listed in the Appendix A.

The p53 system is simulated by deterministic and stochastic methods in this work. The deterministic method regards the interaction of proteins in the p53 system as a set of differential equations. This method can be used to study the average values of protein kinetics after irradiation. The equations are solved by Matlab software. It is noted that the deterministic method is not suitable for the simulation of apoptosis and its related effects because of the state jumps induced by apoptosis. Therefore, the stochastic method is used to simulate apoptosis and the stochastic kinetics of proteins. A stochastic simulating code using the Gillespie method and C++ language is developed for this simulation [46].

### 3.6. Simulation of Cell Response after Exposure to Low- and High-LET Radiation

Lesions induced by low and high LET differ greatly in distribution and complexity, which may result in different consequences. One of the aims of this study is to explore how low- and high-LET radiation affects repair kinetics, respectively, and how the kinetics influences the biological effects. We perform simulations with coupling models of the repair module and p53 system module for radiation with different doses and LETs. The dose is set to be 2.0, 4.0, and 8.0 Gy, and the fraction of the slow compartment is set to be 20%, 40%, 60%, and 80% to represent the repair process after exposure to low- and high-LET radiation.

The activation of the p53 system is initialized by the activation of ATM induced by DSBs. The activation of ATM by DSBs is modeled by a Hill function [15] described in Equation (4).


(4)
$$\frac{1}{\mathrm{AT}M_{0}}\frac{{\mathrm{dATM}}_{\mathrm{act}}}{\mathrm{dt}}=k\frac{DSB^{2}}{DSB^{2}+M^{2}}$$


*ATM*_0_ is the number of inactive ATMs; *ATM_act_* is the number of activated ATMs; *k* is a coefficient; *DSB* is the number of double-strand breaks; and *M* is the half-saturation threshold of the Hill function, which is equal to the number of DSBs when the activation of ATM is half-saturated.

The activation of ATM is the key process of the system, and M is the most important parameter in the Hill function. We change the values of M equivalent to 0.14 Gy [15] and 0.5 Gy [26] to explore its influence on the biological effects. For each condition, 100 cells are simulated to calculate the percentage of apoptosis.

## 4. Conclusions

In the present study, a repair module within the NASIC program has been improved by adding a model with factors regulating slow and fast compartments and corresponding repair kinetics. Results of the repair compartments and complex DSBs showed that a complex DSB is repaired by the resection-dependent compartment. The simulated curves of repair showed that more complex DSBs led to slow repair. Our model indicated that high-LET radiation induces more complex DSBs, thus resulting in a larger fraction of the slow repair compartment.

A model of the p53 system is introduced in the present work to study the dynamic changes in proteins and the fate of cells after radiation. Simulated results showed that the repair kinetics also obviously affected the fate of cells. A higher dose or larger fraction of the slow repair compartment will result in a longer activation of the p53 signal, which leads to a longer cell cycle arrest, more apoptosis, and more secreting of TGFβ. High-LET radiation induces more complex DSBs, resulting in more serious consequences, as aforementioned. The results of the p53 system indicated that the kinetics of DSB repair also influenced the biological effect greatly besides the number of DSBs. More details of DNA repair, cell responses related to p53, and the interplay between these two processes can be obtained by this simulation coupling the repair modules and the p53 system module. The p53 system module can be used to study the effect of radiation quality, other factors on the cell response, and the fate after radiation exposure. However, the module is limited by the complexity of the cellular system and the uncertainty of the parameters, which will be further investigated.

The integrated modules of DSB repair and the model of the p53 system in this study expand the functions of the NASIC biophysical simulation program. These additional modules enable NAISC code to be a powerful tool for analysis of the mechanism of radiobiology from the simulation of physical interactions, chemical reactions, and DNA repair to the cellular response.

## Figures and Tables

**Figure 1 ijms-23-11323-f001:**
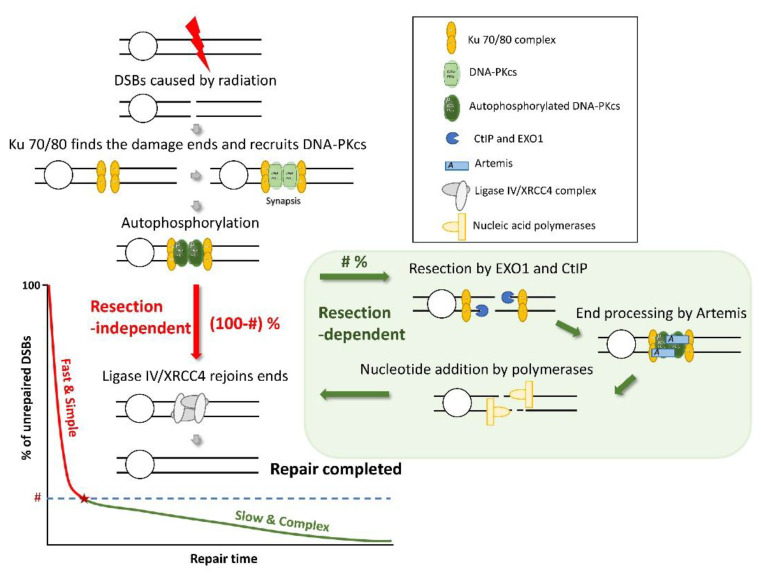
Scheme of the NHEJ repair pathway, including the resection-dependent compartment and the resection-independent compartment. # refers to the percentage of the resection-dependent compartment. Red and blue arrows respectively refer to the steps of the fast and slow processes, and the gray ones have shared steps.

**Figure 2 ijms-23-11323-f002:**
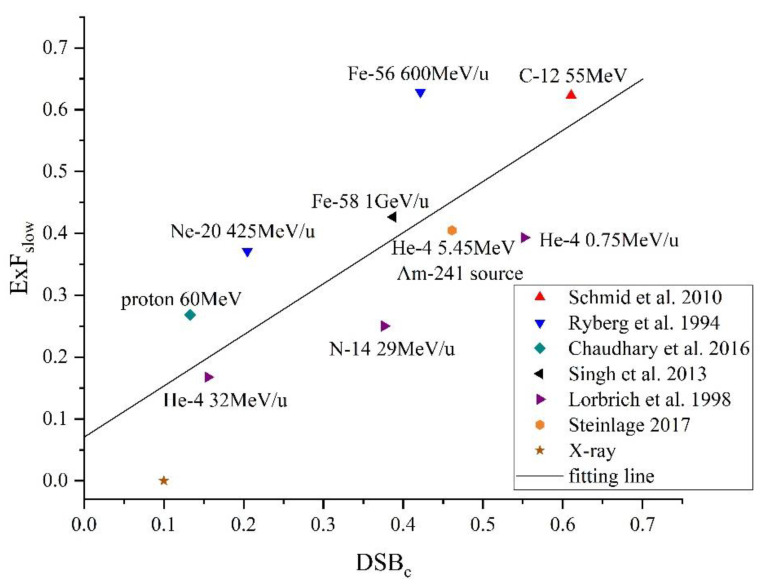
Data and the fitting line of complex DSBs (DSBc) and additional fractions of the slow compartment attributed to complex DSBs (ExF_slow_) [17,29,30,31].

**Figure 3 ijms-23-11323-f003:**
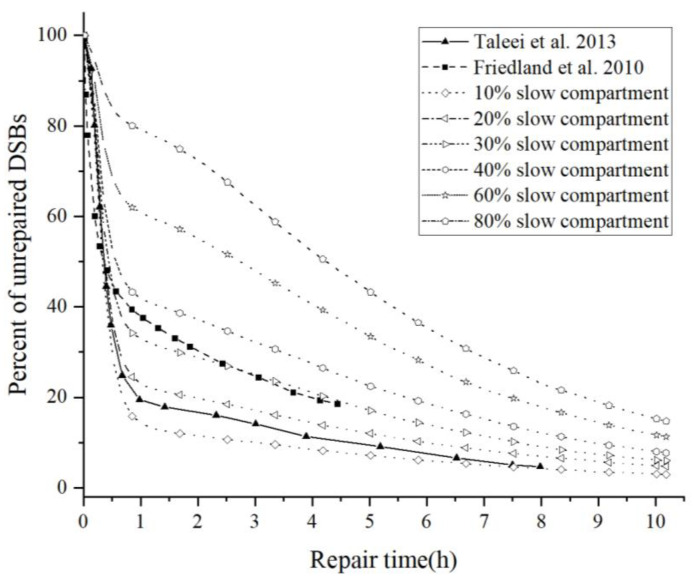
Repair curves of the repair models [21,33].

**Figure 4 ijms-23-11323-f004:**
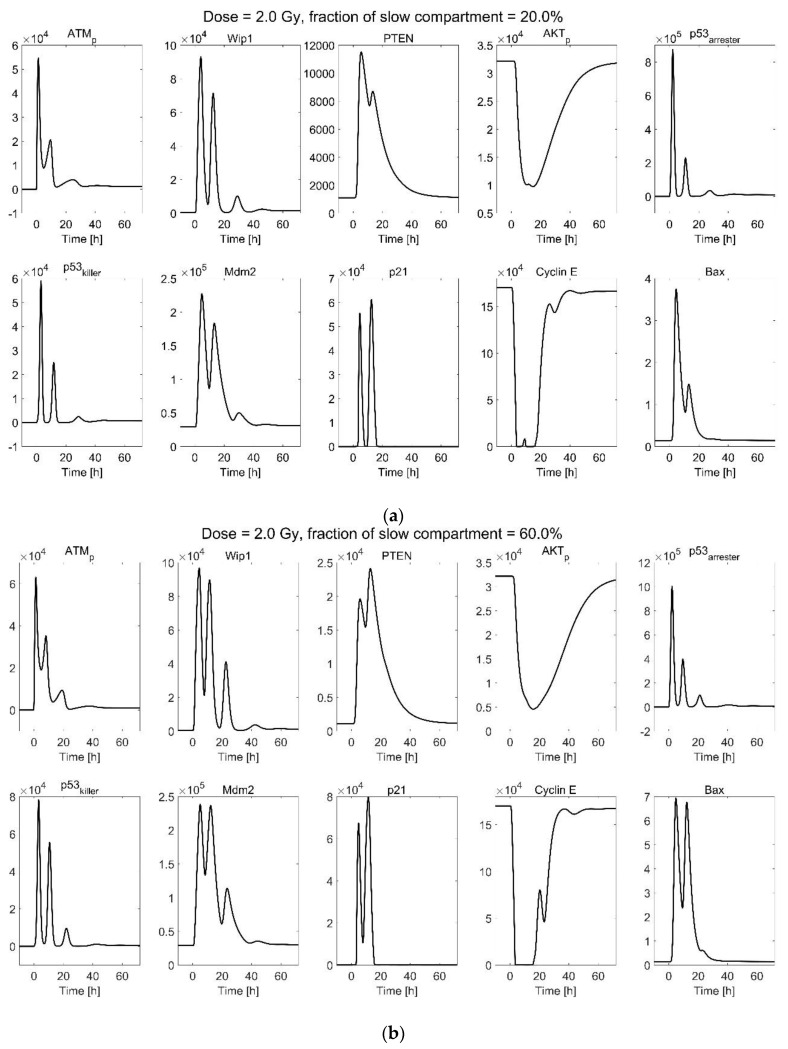
Deterministic simulation trajectories of the p53 system in response to (**a**) 2 Gy irradiation, 20% slow compartment; (**b**) 2 Gy irradiation, 60% slow compartment; (**c**) 8 Gy irradiation, 20% slow compartment; and (**d**) 8 Gy irradiation, 60% slow compartment. ATM_p_ to Bax are proteins related to the p53 system; descriptions of them are listed in the Appendix A.

**Figure 5 ijms-23-11323-f005:**
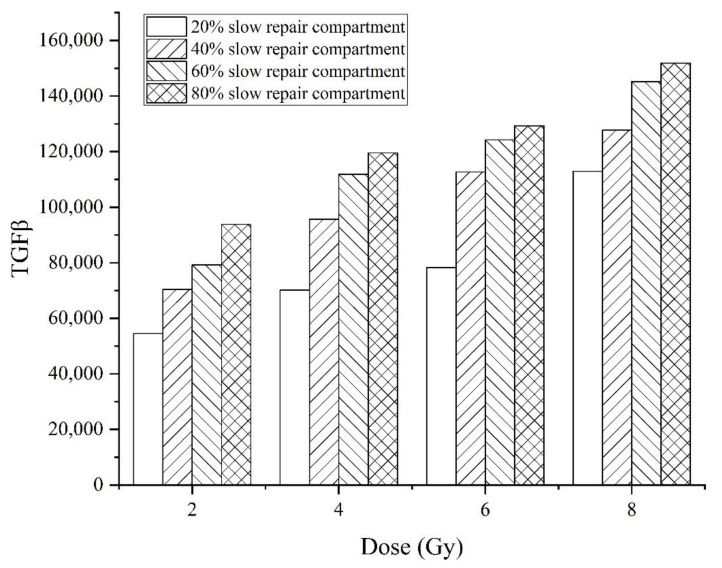
TGFβ secreting in the conditions of exposure to 2–8 Gy dose irradiation and repaired by 20–80% slow repair compartment.

**Figure 6 ijms-23-11323-f006:**
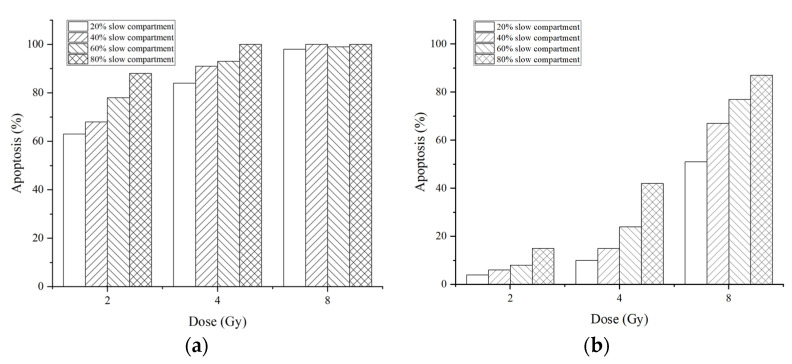
Percentages of apoptosis for cells with the half-saturation threshold of ATM equivalent to 0.14 Gy (**a**) and 0.5 Gy (**b**) at 72 h after 2.0–8.0 Gy irradiation. The cells are repaired by 20–80% slow compartments.

**Figure 7 ijms-23-11323-f007:**
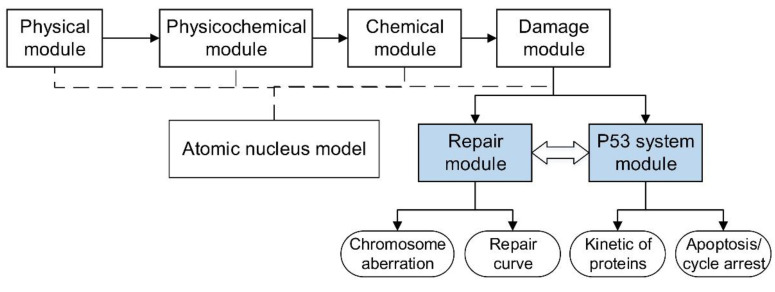
Scheme of the simulation modules within NASIC. The new modules added in this study are filled with blue shading.

**Figure 8 ijms-23-11323-f008:**
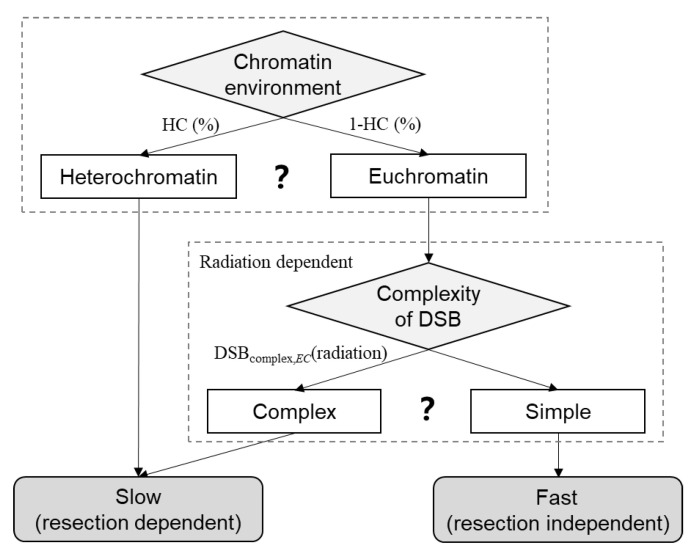
Scheme of the factors regulating the slow and fast repair compartments. HC is the fraction of heterochromatin. DSB_complex,EC_(radiation) is the fraction of complex DSB in the DSB located in euchromatin.

**Figure 9 ijms-23-11323-f009:**
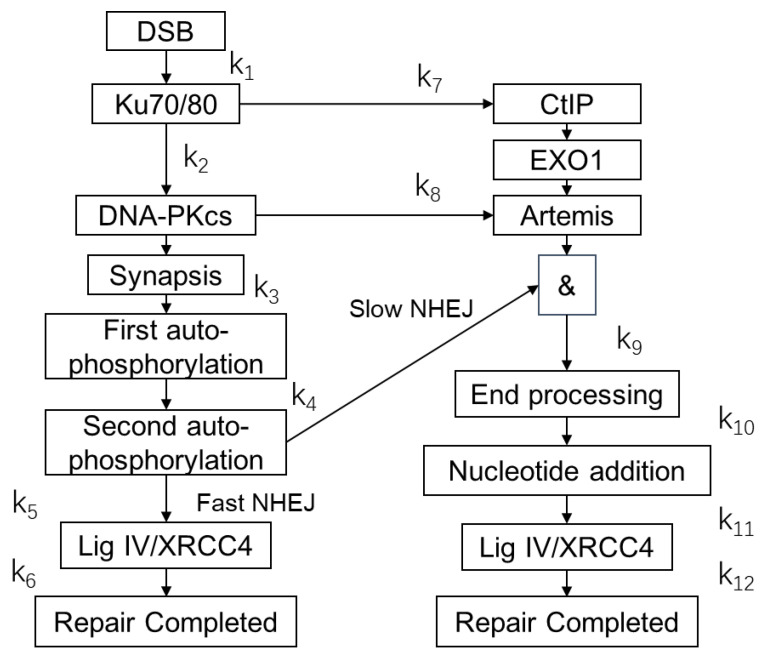
Scheme of the DNA repair model with fast and slow compartments. Ku70/80, DNA-PKcs, CtIP, EXO1, Lig IV, XRCC4, and Artemis are proteins attending NHEJ. The k_1_ to k_12_ are rate constants of repairing processes listed in Table 1.

**Figure 10 ijms-23-11323-f010:**
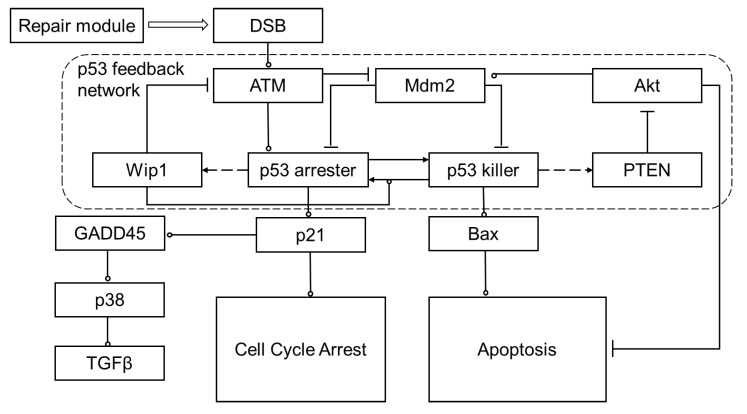
Scheme of the p53 system and related modules of cell cycle arrest apoptosis and TGFβ secreting. Arrow-headed dashed lines indicate positive transcriptional regulation, arrow-headed solid lines—protein transformation, circle-headed solid lines—positive influence or activation, hammer-headed solid lines—inhibitory regulation. The description of the proteins, genes, and RNAs are listed in the Appendix A.

## Data Availability

The data presented in this study are available in the article and Appendix A.

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
