# Peer review of "Modeling of DNA Damage Repair and Cell Response in Relation to p53 System Exposed to Ionizing Radiation"

_ijms, 2022, doi:10.3390/ijms231911323_

Round 1

Reviewer 1 Report

This manuscript describes an interesting and potentially useful effort to use Monte Carlo simulations to model the effects of various levels and types of radiation on the generation and repair of double strand breaks in human cells as well as apoptosis. The study seems to provide information that is within the realm of plausibility. The parameters added to the existing NASIC Monte Carlo code are reasonable ones to add.

            That said, it is difficult to assess the potential for this approach. This is primarily because the study is not supported by detailed comparisons with data obtained from living cells to determine how closely the simulations track real life. The omission of such data greatly limits the impact of this work.

On many pages, it looks like line numbers became intermingled with the text.

In many places, “kinetic” should be changed to “kinetics”.

There are quite a few statements made without attribution to back them up. For example, at the top of p2, it is stated: “Compared to low LET radi-ation, high LET radiation induces more complex DSBs. The kinetics of DSBs repair differ greatly between low and high LET radiation, which influence the p53 system, leading to different results.”  Citations are needed here. Also, what is meant by “more complex DSBs”? How are they more complex?

Author Response

This manuscript describes an interesting and potentially useful effort to use Monte Carlo simulations to model the effects of various levels and types of radiation on the generation and repair of double strand breaks in human cells as well as apoptosis. The study seems to provide information that is within the realm of plausibility. The parameters added to the existing NASIC Monte Carlo code are reasonable ones to add.

That said, it is difficult to assess the potential for this approach. This is primarily because the study is not supported by detailed comparisons with data obtained from living cells to determine how closely the simulations track real life. The omission of such data greatly limits the impact of this work.

Reply: Thanks for your comments and suggestion. We agree that the validation of models is very important for the potential applications of this model. We tried to compare our simulation results with experiments. Results show that our model is consistent with experiments in some aspects. The DNA repair curves agree with the results of Taleei et al. and Friedland et al. which agreed with repair curves obtained by cellular experiments. Our results of protein dynamics in the p53 system reflect the oscillation of the p53 system.

However, we cannot perform further validation because of some insurmountable problems. The difficulty in simulating the real conditions is a common challenge in modeling the biological system because of the complexity of the system and the uncertainties of parameters. Even though a direct comparison between simulation results and experimental results is the best way to evaluate the model, it is difficult to compare the results with experiments directly because of the complexity of the cell. We take the DNA repair curves as examples. There are nonnegligible divergences between Taleei et al. and Friedland et al. although they both aimed to describe the process of DNA repair and their parameters were both from experiments. The main aim of the modeling study is to provide a tool to explore how a factor influences the cell, for example, the influence of the complex DSBs on DNA repair and the p53 system. The model is limited by the complex cellular system and uncertainty of parameters. Despite the limitations, the model of the cellular such as models established by Ma et al. (PNAS 2005, 102, 14266-71), Dolan et al. (PLoS Comput Biol 2015, 11, (5), e1004246) and Hat et al. (PLoS Comput Biol 2016, 12, (2), e1004787.) are developed to study the interactions of proteins and exploring the influencing factors. We notice that the simulation results of proteins dynamic were not compared with cell experiments directly in these papers for validation. We think that it is also because these models can hardly be validated by direct comparison with the experiments. However, these models provide effective tools to study cell response and fate systematically.

Although our model is limited and the model cannot be used to precisely predict the results by giving the conditions of the experiments, it can be used to study the effect of radiation quality and other factors such as siRNA or agents on the DNA repair and cellular response. In the future study, we will design experiments by controlling the factors to validate our model. Moreover, we can choose more accurate parameters of the model in the future with the development of quantitative experiments.

To make these points clearer, we added some discussions in the conclusion:

“The p53 system module can be used to study the effect of radiation quality, other factors on the cell response and the fate after exposure to radiation. However, the module is limited by the complexity of the cellular system and the uncertainty of the parameters, which will be further investigated.”

We also added some sentences in the introduction to illustrate the aim and limitation of modeling the p53 system:

“The main aim of modeling the p53 system is to explore how the related factors influence the cell response and cell fate by taking into account the complexity and uncertainty of the cellular system.”

On many pages, it looks like line numbers became intermingled with the text.

Reply: The revised version is composed by using the template of the journal. We carefully check the manuscript to avoid this fault.

In many places, “kinetic” should be changed to “kinetics”.

Reply: we correct them and carefully check again,

There are quite a few statements made without attribution to back them up. For example, at the top of p2, it is stated: “Compared to low LET radiation, high LET radiation induces more complex DSBs. The kinetics of DSBs repair differ greatly between low and high LET radiation, which influence the p53 system, leading to different results.”  Citations are needed here. Also, what is meant by “more complex DSBs”? How are they more complex?

Reply: Our statement is not quite appropriate for the example at the top of p2 resulting in misleading. The statement “more complex DSBs” means a larger number of complex DSBs rather than “more complex”. We correct it. We added citations as support and rephrase the sentences to:

“The system of p53 consists of several feedback loops [14, 15], which means that the kinetics of the external factors that activate the p53 system also influence the p53 system [16]. Besides the number of DNA lesions, the p53 system is influenced by the dynamics of DNA lesions [16]. Compared to low LET radiation, high LET radiation induces a larger number of complex DSBs [9, 17]. Because of different fractions of complex DSBs, the kinetics of DSBs repair differ greatly between low and high LET radiation [17], which influences the p53 system, leading to different results.”

We check the whole manuscript to revise the statements. If the statement is supported by the literature, we attached citations, other statements without citations are supported by the results in this manuscript.

Moreover, we added a figure and some sentences in the introduction section to further illustrate the DNA repair process based on the suggestion of the other reviewer.

Reviewer 2 Report

The article offers mechanistic insights into radiobiological simulation of ionizing radiation impact on the repair pathways of complex DNA lesions (here, DSBs) and on cell response modulated by p53 system. The authors present interesting observations, and the article is well-written. The introduction was comprehensive and informative. Results are clearly described and illustrated. The discussion is satisfactory.

I recommend the article for publication after minor corrections.

-        Figures: please add descriptions of abbreviations in the figure captions

-        line 135: please provide exact radiation conditions from literature

-        methods could be described with more detailed description of all parameters

-        introduction would benefit from the graphical figure presenting the scheme of NHEJ pathway, its steps and proteins taking part, showing the fast and the slow pathway

Author Response

The article offers mechanistic insights into radiobiological simulation of ionizing radiation impact on the repair pathways of complex DNA lesions (here, DSBs) and on cell response modulated by p53 system. The authors present interesting observations, and the article is well-written. The introduction was comprehensive and informative. Results are clearly described and illustrated. The discussion is satisfactory.

Reply: Thanks for your comments and suggestions.

I recommend the article for publication after minor corrections.

-        Figures: please add descriptions of abbreviations in the figure captions

Reply: We added necessary descriptions of abbreviations in the figure captions. But for the abbreviations of proteins and genes, we marked the citations to find out the description because the number is too large.

-        line 135: please provide exact radiation conditions from literature

Reply: We think it is better to describe the radiation conditions near the data points in the figure in the results (section 3,1, figure 6 in the revised version). We add the sentences in line 135:

“These experiments cover the cells irradiated by X-ray [16], proton [33], helium ions [28, 34], carbon ion [16], nitrogen ion [34], neon ion [35] and iron ion [35, 36]. The type and energy of the particles for each condition are marked in the figure in section 3.1.”

-        methods could be described with more detailed description of all parameters

Reply: We added descriptions of parameters in the materials and methods.

We add the sentences in section 2.4:

“The parameters k1 – k12 are the rate constants of protein recruitments. The k is the reciprocal of the characteristic time of the recruitment. If the distance between two DNA ends is less than a value, the synaptic complex is formed. The value is defined as syn-apsis distance. The diffusion coefficients of DNA ends are obtained from Friedland et al. [22].”

And added the sentences in section 2.5:

“The interactions between proteins, mRNA or genes are modeled and described as chemical reaction formulas. The rate constants of these interactions are listed in supplement materials table S2 and S3.”

We added the description of p53 arrester:

“Then activated ATM stabilizes and activates p53 by phosphorylating it to one of its active phosphoform (at Ser15 and Ser20), p53 arrester.”

-        introduction would benefit from the graphical figure presenting the scheme of NHEJ pathway, its steps and proteins taking part, showing the fast and the slow pathway

Reply: We add a figure in the introduction to briefly describe the scheme of NHEJ including the resection-dependent (slow) compartment and the resection-independent (fast) compartment. The detailed scheme of the NHEJ is shown in section 2.4.
